# A Methodology for Controlling Bias and Fairness in Synthetic Data Generation

Enrico Barbierato [1,*], Marco L. Della Vedova [2], Daniele Tessera [1], Daniele Toti [1] and Nicola Vanoli [1]

1 Faculty of Mathematical, Physical and Natural Sciences, Catholic University of the Sacred Heart,
25121 Brescia, Italy; daniele.tessera@unicatt.it (D.T.); daniele.toti@unicatt.it (D.T.);
nicola.vanoli@unicatt.it (N.V.)

2 Department of Management, Information and Production Engineering, University of Bergamo,
24129 Bergamo, Italy; marco.dellavedova@unibg.it

* Correspondence: enrico.barbierato@unicatt.it

**Abstract:** The development of algorithms, based on machine learning techniques, supporting (or even replacing) human judgment must take into account concepts such as data bias and fairness. Though scientific literature proposes numerous techniques to detect and evaluate these problems, less attention has been dedicated to methods generating intentionally biased datasets, which could be used by data scientists to develop and validate unbiased and fair decision-making algorithms. To this end, this paper presents a novel method to generate a synthetic dataset, where bias can be modeled by using a probabilistic network exploiting structural equation modeling. The proposed methodology has been validated on a simple dataset to highlight the impact of tuning parameters on bias and fairness, as well as on a more realistic example based on a loan approval status dataset. In particular, this methodology requires a limited number of parameters compared to other techniques for generating datasets with a controlled amount of bias and fairness.

**Keywords:** bias; fairness; structural equation modeling; data generation; machine learning

## 1. Introduction

According to Forbes [1], our world has never been more interconnected; amazing leaps in technology have made a global community feel local in scale. As the world becomes more and more data-driven, smart businesses look to fully realize the benefits of the data revolution (not without perils [2]), from streamlining internal processes and communicating more ably with current and potential customers to lowering costs and creating jobs. In most of the cases, industries use data to train and develop AI models able to extrapolate patterns and information that would otherwise be impossible to find. Developing successful AI and machine learning models requires access to large amounts of high-quality data [3]. However, collecting such information is a challenging task because: (i) some types of data are costly to collect, or they are rare; for example, collecting data representing the variety of real-world road events for an autonomous vehicle may be prohibitively expensive, etc.; (ii) many business problems that are solved through AI/ML models require access to sensitive customer data, such as medical or financial.

Generating synthetic data that reflect the important statistical properties of the underlying real-world does not, however, solve all of the problems; in fact, in most cases, real data contain some type of bias that has to be removed or at least handled before the training process. According to the Collins English Dictionary, the term bias denotes "a tendency to prefer one person or thing to another, and to favor that person or thing". Though this definition may seem relatively unambiguous, it may indeed be used with a number of different meanings in accordance with the object (person/thing) of such a bias; as such, this concept cannot be directly applied to a software application. Not only can bias affect different types of data, from demographic to behavioral, but it can also be found in text,

images and even language. Its actual measurement is a rather complex task due to the fact that bias may be poorly defined from a mathematical point of view.

In this sense, the contribution of this work consists of presenting a methodology to control the amount of bias and fairness that may lie within a dataset. The term "bias" is therefore used in the present work to indicate the influence that certain data elements or variables may have upon other elements in a given dataset, and, thus, how the latter are affected by the former in terms of direct of indirect correlations. In this regard, the approach proposed in this work allows a user to exactly know and set a priori the amount of bias and the strength of correlations among variables. Moreover, even though this methodology is a way to automatically generate synthetic data from a structural equation modeling (SEM) representation, it is not domain-specific, and thus may in principle be applied and used in different contexts or scenarios.

The rest of the article is organized as follows. In Section 2, the open-source projects and tools that are currently available for the generation of new data are presented. Section 3 introduces and explains the operational aspects of the presented methodology and then compares it with a Bayesian network. Section 4 shows how the proposed methodology enables the generation of multiple datasets sharing similar features but having a different amount of bias each. Finally, in Section 5 conclusions are drawn.

## 2. Related Work

Given the wide adoption of automated prediction systems in the last few years, being able to develop a fairness-aware predictive algorithm has become of fundamental importance. Friedman et al. [4] cite a case of a software system for booking airline tickets favoring two specific airlines over the others, making the ranking process biased. Although, in this case, the problem can be easily identified and is relatively circumscribed, more complex scenarios present a widespread bias, which is rather difficult to detect and explain. In particular, the authors define bias as "computer systems that systematically and unfairly discriminate against certain individuals or groups of individuals in favor of others", building the semantic foundation based on different definitions: (i) preexisting bias, which is embedded in culture and habits and reflects individuals and our society; (ii) technical bias, depending on the hardware capabilities of the machine or on the deployed algorithms, including cases where the generation of random numbers is not correct; (iii) emergent bias, which typically involves changes in the habits of the system users, both in terms of new aspects of the society and mutated cultural values. Many studies focus on detecting and removing the bias from existing and widely available datasets. For example, in [5], the authors provide innovative ways to create synthetic datasets from existing ones; in this case, the new datasets tend to be bias-free and as fair as possible. In [6], the authors identify the cause of bias in the heterogeneity of data. As a result, this can affect the model's performance, significantly decreasing its accuracy.

The fairness problem is currently addressed by three types of methods: (i) pre-processing techniques, which revise input data to remove information correlated to sensitive attributes; (ii) in-process methods, which add fairness constraints into the model learning process; and (iii) post-process approaches, which adjust model predictions after the model is trained [7]. In particular, the pre-processing algorithms rely on the idea that the dataset is the cause of the discrimination that a machine learning algorithm might learn; thus, the modification of the dataset can help the learning algorithm to remain bias-free.

Some of these pre-processing techniques include methods for modifying data such as massaging, which consists of changing the labels of some observations in the data, *reweighting*, which assigns weights to individuals in order to balance the data, and sampling, which varies the sample sizes of subgroups in the dataset to remove the discrimination [8]. Recently, new studies have shown the possibility of using generative adversarial networks (GAN) to remove discrimination. For example, in [5], authors develop a new GAN model that is free from disparate treatment and impact in terms of the real protected attribute,

while retaining high data utility (reflecting the fact that the generated data should preserve the general relationship between attributes and decision in the real data).

While all of these methods rely on existing datasets as a starting point, in order to study the amount of bias and fairness learned by the trained algorithms, it is the authors' conviction that researchers might find it useful to have datasets at their disposal where all correlations, biases and distributions of variables are known a priori. The pre-processing approach presented in this work provides the possibility of using a training dataset that is fully and completely understood by the user.

With reference to synthetic data generation, SynSis [9] uses a method for training a hidden Markov model (HMM) over smart home data corresponding to the movements taken by the residents to produce a sequence of activities. The next phase consists of training a dedicated HMM for a specific activity; the last step relates to the regression learners training creating extra timestamp-based data. SynSis was tested against related data by calculating the Euclidean distance over different time periods.

Synthetic data are often used to learn large size convolutional neural network (CNN) models [10], while Jaderberg et al. [11] developed an end-to end system meant to recognize text in an image by using a CNN trained on synthetic data (made up of nine million, $32 \times 100$ pixels images), generated by a mixture of random choices concerning the character font and the border/shadow rendering, and k-means clustering regarding the base coloring.

A data augmentation technique based on GANs to create synthetic medical images starting from real data [12] is used by the authors to report a significant increase in sensitivity and specificity. The idea that synthetic data augmentation can support deflating inherent bias in large-scale image datasets is presented by Jaipuria et al. [13] as an approach consisting of mixing GAN and gaming-engine simulations, creating semantically consistent data of targeted task-specific scenarios [13]. Similarly, a GAN trained over six experiments with a mix of numerical and categorical variables originated from three datasets is discussed by Arvanitis et al. [14]. The generated synthetic dataset was validated across correlation matrices of real and generated data by using the Jaccard similarity.

Techniques based on evolutionary algorithms can be deployed to generate training and testing scenarios (see Lou et al. [15,16]). Furthermore, Shand et al. [17] discuss a data generator called HAWKS, defined over an evolutionary algorithm to evolve the cluster structure of a synthetic data set. While HAWKS focuses on clustering scenarios, the methodology proposed in the present work aims to be more general and independent of ML paradigms. Furthermore, HAWKS produces synthetic data clusters (quite interestingly, the authors compare the output to blobs generated by Python's sci-kit learn), although the notion of bias is not really taken into account (nor is the focus of the reviewed approaches), differently from the technique proposed in this work.

Aside from the scientific literature, data augmentation is a topic covered by commercial tools, such as Synthesized (https://www.synthesized.io/), which provides a Python library measuring the extent of bias in a dataset, together with reporting facilities, the capability to flag sensitive variables and fairness scoring tools. Clearbox AI (https://app.clearbox.ai) is a data augmentation tool performing anonymization and reducing overfitting and imbalances starting from an original dataset. DataGen (https://www.ceadar.ie/pages/datagen/) offers the capability to generate datasets according to a set of features defined by the user; however, when the process is automatically executed, it must be based on an existing dataset. Manual generation is made possible as the user can specify in advance a set of relationships among the dataset variables.

## 3. Methodology for Data Generation

This section presents the methodology for the generation of datasets to be used in machine learning applications. The objective is to generate a dataset $\mathcal{D}$ consisting of $n$ records described by $m$ categorical features, i.e., features with a finite set of values. In turn, each feature $j$ is characterized by $q_j$ values.

The proposed methodology is based on the definition of a probabilistic network that accounts for the dependencies among data features, while appropriate samplings of this network will be used to generate the datasets. More precisely, a five-step approach was applied, consisting of:

1.  Defining the probabilistic network that characterizes the initial dependencies and their strength, among features;
2.  Altering the bias on the dataset by tuning the direct influence among attribute pairs, as well as the overall amount of bias of a specific attribute;
3.  Deriving the multivariate probabilistic distribution that summarizes this probabilistic network;
4.  Sampling the multivariate normal distribution;
5.  Converting the samplings into a dataset with categorical features.

As already discussed in Section 2, most of the state-of-the-art, non-proprietary approaches for generating or enlarging datasets rely on both existing data and large conditional probability tables in order to address dependencies among the various data features. In contrast, the proposed methodology generates new datasets using only a limited number of parameters to describe the strength of ties among the various data features. More precisely, similarly to Bayesian networks, this probabilistic network is a weighted directed acyclic graph (DAG) whose nodes represent feature variables and whose edges describe the direct probabilistic influences. The weights of the edges summarize the strength of the corresponding influence. The proposed methodology is based on a structural equation modeling (SEM) framework—see, e.g., [18,19]—to model the dependencies among features. More precisely, SEM representations are usually applied, in combination with factor and multiple regression analyses, to model datasets with sets of equations and structures. This approach also accounts for data variability, noise and discrepancy by using latent variables. In contrast with usual SEM applications, the methodology described in this work uses SEM representations as the starting point for data generation. Indeed, each feature is associated with a latent Gaussian random variable that, in turn, is represented by a node in the DAG. The values of the features are computed from sampling the joint distribution of all of the latent variables.

As a running example, let $\mathcal{D}$ be a dataset characterized by $m = 4$ features, namely $A$, $B$, $C$ and $Z$. Given $A$, $B$ and $C$ as independent features, the purpose is to model their dependencies on the feature $Z$, that is, to derive the conditional probability distribution $P(Z|A, B, C)$. Figure 1 depicts the SEM representation of these features, where $X_A$, $X_B$, $X_C$ and $X_Z$ are the latent random variables associated with the considered features. Moreover, the edge weights $\alpha_A$, $\alpha_B$ and $\alpha_C$, i.e., the regression coefficients in SEM terminology, summarize the strength of the dependencies among the corresponding pair of random variables.

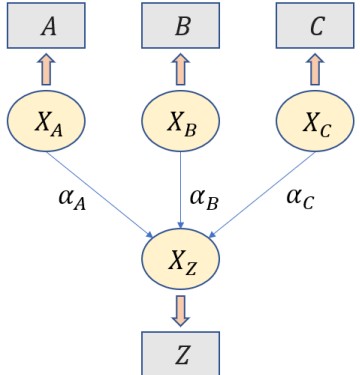

**Figure 1.** Example of a SEM representation describing a dependent feature $Z$ as a function of three independent features, namely $A$, $B$ and $C$. Rounded nodes represent the latent variables, whereas squared boxes denote their corresponding features.

The building process of this SEM starts by initializing each latent random variable $X_J$, for each $J \in \{A, B, C, Z\}$, with a Gaussian noise $\mathcal{N}(0, \sigma_J^2)$, with zero mean and variance equal to a given $\sigma_J^2$ that represents the conditional variance of the latent variable $X_J$. Although there is not any theoretical restriction that prevents the initialization of latent variables with arbitrary probability distributions, as discussed later in this section, using Gaussian distributions simplifies the algebraic composition of random variables. Moreover, since this methodology focuses on generating categorical features, the desired marginal probabilities associated with the feature values are imposed by the discretization process of the composition of latent variables, regardless of the type of probability distribution used for their initialization. The following step addresses the dependencies among random variables. These dependencies are accounted for by adding to the destination random variable a weighted sum of the conditional independent parent random variables. In the examples proposed, this sum is equal to:

$$X_Z = \alpha_A X_A + \alpha_B X_B + \alpha_C X_C + \mathcal{N}(0, \sigma_Z^2)$$

where $\alpha_A$, $\alpha_B$ and $\alpha_C$ are the regression coefficients associated with the $(X_A, X_Z)$, $(X_B, X_Z)$ and $(X_C, X_Z)$ edges, respectively. Since $Z$ is a weighted sum of random variables, its expected value is the weighted sum of the their expected values. In general, given a random variable $Y$, such that:

$$Y = \alpha_1 X_1 + \alpha_2 X_2 + \ldots + \alpha_n X_n$$

and the variance of $Y$ is equal to:

$$\sigma_Y^2 = \sum_{i=1}^{n} \alpha_i^2 \sigma_{X_i}^2 + 2 \sum_{j=1+n}^{n} \text{Cov}(X_i, X_j)$$

In the example proposed, $A$, $B$ and $C$ are independent and Gaussian distributed with a mean value equal to zero. Hence, the covariance between the independent variables $X_i$ with $i \in A, B, C$ is zero and the variance $\sigma_Z^2$ of $X_Z$ is computed as:

$$\sigma_Z^2 = \alpha_A^2 \sigma_A^2 + \alpha_B^2 \sigma_B^2 + \alpha_C^2 \sigma_C^2 + \sigma_Z^2$$

All dependencies among these variables are accounted for by a multivariate Gaussian distribution $\mathcal{N}(\bar{0}, \Sigma)$, where $\bar{0}$ is the $m$-dimensional zero vector representing all of the zero mean values and $\Sigma$ is the variance–covariance matrix of the $m$ latent variables. The elements $\Sigma_{i,j}$ of this matrix are the covariances between variables $i$ and $j$ and are computed as $\Sigma_{i,j} = \mathbb{E}[(X_i - \mathbb{E}[X_i])(X_j - \mathbb{E}[X_j])]$. The presented methodology exploits the property that $\Sigma$ can be derived from the $m \times m$ adjacency matrix $A$ describing the probabilistic network. In detail, each $A_{ij}$ element of $A$ is equal to the weight of the edge $(i, j)$ between nodes $i$ and $j$ and $A_{ij} = 0$ if there is not any edge between the nodes. In addition, let $S$ be the diagonal $m \times m$ matrix whose elements $S_{ii}$ are the conditional variances $\sigma_i^2$ of the corresponding latent variables $\{X_j; j \in [1, \ldots, m]\}$. It is then possible to derive $\Sigma$ as:

$$\Sigma = QSQ^T$$

where $Q = (I - A^T)^{-1}$, where $I$ is the $m \times m$ identity matrix.

Once the probabilistic network is fully described by the multivariate Gaussian distribution $\mathcal{N}(\bar{0}, \Sigma)$, the last two steps of our methodology aim to generate the dataset composed of the $n$ requested observations. More precisely, it is necessary to initially derive the latent dataset composed of $n$ samples of $\mathcal{N}(\bar{0}, \Sigma)$. As proposed in [20], the starting point is to compute, by applying the Cholesky factorization, a matrix $B$ such that $BB^T = \Sigma$. Hence, numerical sampling $x_i \in \mathbb{R}^m; 1 \leq i \leq n$, of $\mathcal{N}(\bar{0}, \Sigma)$ can be derived as $x_i = \mu + B z_i$, where

$\mu = \bar{0}$ is the $m$ dimensional null vector, since all $m$ expected values are equal to zero, and $z_i$ are independent samplings of a univariate standard normal distribution $\mathcal{N}(0,1)$.

The last step of the proposed methodology focuses on deriving the final dataset where, as previously introduced, each record is described by categorical features. For such a purpose, the latent dataset has to be converted into a per-feature finite set of ordinal values by applying an appropriate data binning. Since each feature $K$ ranges among $q_K$ values, $q_K - 1$ thresholds, named latent variable cutoffs, have to be identified. This discretization process has some implications on the bias among the feature values. More precisely, latent variables, described by Gaussian probability distributions, have an implicit ordering on their samples, which, in turn, results in an ordering among the categorical values of their binned features. Although the ordering among the categorical values of each feature can be arbitrarily imposed (e.g., gender, civil/marital status and living area), the impact of the dependencies of the other latent variables is determined by the position of its bin, that is, by its corresponding cutoffs. Indeed, since the latent variable of each node is derived as the weighted sum of the internal Gaussian distribution and the latent variables of connected nodes, large edge weights increase the probabilities of large values, and, hence, the frequencies of outer bins. Notably, the partitioning of the probability density function of a latent random variable $X_k$ in $q_K$ areas with the same size results in a balanced dataset with respect to the values of this feature. Conversely, a dataset having features with unbalanced marginal frequencies can be generated by specifying appropriate cutoffs associated with uneven probability distribution areas. In detail, the proposed methodology leverages the fact that each latent random variable part of the $\mathcal{N}(\bar{0}, \Sigma)$ Gaussian is itself a Gaussian distribution with mean equal to zero. The cutoffs are then computed such that the probability associated with each bin matches the requested marginal probability.

Once these cutoffs have been determined, the latent dataset is converted into the feature domain dataset $\mathcal{D}$ by bucketing each latent random variable accordingly. Figure 2 shows an example of a latent variable converted into a balanced ordinal random variable with three possible categorical values.

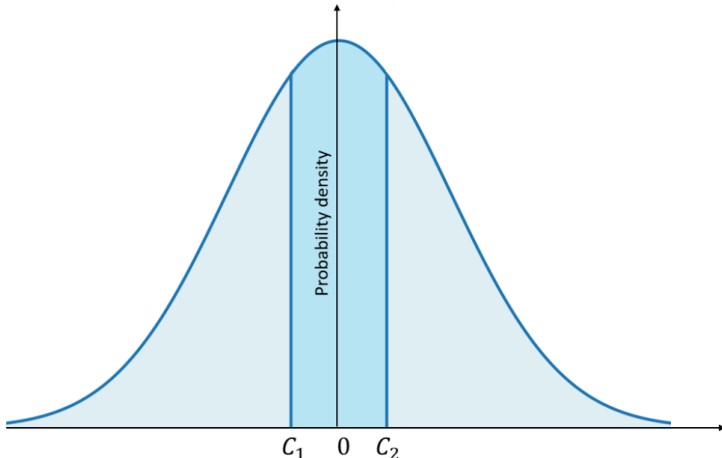

**Figure 2.** Probability density function of a latent random variable to be converted into a three-value discrete random variable. The two cutoffs, namely $c_1$ and $c_2$, split area under the curve into three regions associated with three feature outcomes with the same probability.

Note that, if latent variables $X_i$ are initialized with a non-Gaussian distribution, then dependencies among the various features are not described any more by $\mathcal{N}(\bar{0}, \Sigma)$. This is because, in general, the algebraic composition of non-Gaussian random variables has an arbitrary probability distribution. In this case, the latent dataset can be derived by sampling this arbitrary distribution via numerical techniques.

### 3.1. Comparison with Bayesian Networks

Before analyzing the results achievable through the proposed methodology, it is interesting to compare the number of parameters required to generate a dataset by using this approach with respect to the number of parameters needed when using a Bayesian network.

Unlike all data augmentation techniques, the proposed approach does not rely on existing data and therefore cannot be easily compared to these techniques. As a consequence, the data generation method is compared only with Bayesian networks whose conditional probability table (CPT) values are not derived from data, but set as tuning parameters.

The network depicted in Figure 3 generates a dataset containing financial and personal information to be used for evaluating the possibility on whether to grant a loan or not by a machine learning algorithm. In this scenario, variables such as "Economic Status", "Credit History", "Gender", "Ethnicity" and "Loan Status" can take on two different categorical values, whereas the others can take on three. Note that, not to clutter the presentation, in what follows, each node represents both a latent variable as well as the corresponding categorical feature.

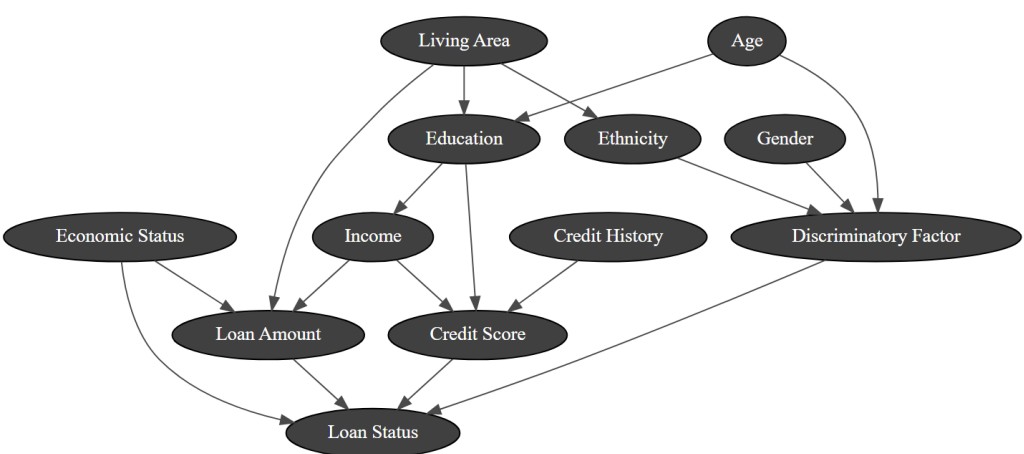

**Figure 3.** Loan approval network: this structure could represent a real scenario that can be used to produce a synthetic dataset.

The first step to build such a dataset using a Bayesian network consists of creating and populating the CPT, which is defined for a set of discrete and mutually dependent random variables to display conditional probabilities of a single variable with respect to the others (i.e., the probability of each possible value for a variable if the values taken on by the other variables are known) [21]. For example, to build from the ground the "Education" variable CPT, it would be necessary to set a value for all possible combinations $p(Education|Age, LivingArea)$ such that $\sum_i p(Education_i|Age, LivingArea) = 1$ holds, where $Education_i$ represents the $i$-th value of the correspondent variable. Variable "Education" is, in fact, conditionally dependent on variables "Age" and "Living Area". Since variables "Education", "Age" and "Living Area" can take on three values each, the total number of parameters needed to fully characterize the "Education" CPT would be 18. If this process is repeated for all variables, a total of 185 would be needed to fully characterize such a Bayesian network.

In contrast to that, the proposed methodology only needs 29 parameters, one for each conditional variance (needed to initialize the latent Gaussian variables), for a total of 12, and 17 for each regression coefficient, that is, the weight of each edge occurring in the network (see Figure 3).

In conclusion, this work highlighted how the proposed methodology drastically lowers the amount of parameters needed to generate synthetic datasets with respect to traditional Bayesian networks, while keeping the final dataset balanced in terms of equally distributed observations.

## 4. Experiments

This section discusses the results obtained via the proposed methodology for two different probabilistic networks: one with a simpler structure, reported in Figure 4, and a more complex one, based on existing datasets and reported in Figure 3.

The first network against which the experiments have been carried out is made up of six nodes and six edges. All of its variables are categorical and generated from latent variables as described in Section 3, and the node $Z$ represents the target node. Nodes $A$, $C$, $D$ and $E$ can assume three different values each, whereas nodes $B$ and $Z$ can only assume two each. As a reminder, the number of nodes, edges and values for each node is arbitrary; to make a comparison, building such a network by manually populating each conditional probability table would require 39 parameters, whereas, by using our method, only 13 are needed.

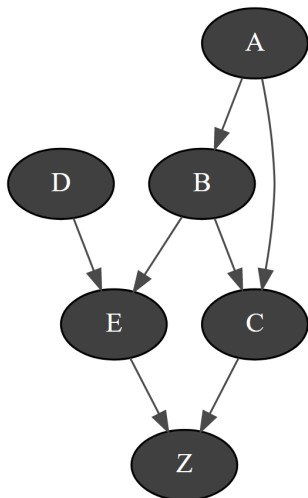

**Figure 4.** Representation of the network used to produce the results.

The remainder of this section is organized as follows. Section 4.1 presents the metrics used for evaluating the bias and fairness in the dataset. Section 4.2 describes the experimental setup. A simple network (Figure 4) is evaluated in Section 4.3 to study the impact of edge weights and in Section 4.4 to study the impact of conditional variances. Finally, a more complex network (Figure 3) is considered in Section 4.5 to show an example of a dataset with controlled discrimination.

### 4.1. Metrics

To compare the impact of variables on the classification node and to study the amount of bias existing in the datasets, two well-established and widely used metrics in the field of data fairness have been proposed, i.e., mutual information (MI) and demographic parity (DP).

Mutual information [18] is a measure of dependence or "mutual dependence" between two random variables, namely $X$ and $Y$. As such, the measure is symmetrical, meaning that $I(X;Y) = I(Y;X)$. In particular, it measures the average reduction in uncertainty about the samples of $X$ that result from learning the samples of $Y$. The mutual information between two discrete random variables $X, Y$ jointly distributed according to $p(x, y)$ is given by:

$$I(X;Y) = \sum_{x,y} p(x,y) log \frac{p(x,y)}{p(x)p(y)}$$

where $x$ and $y$ are the samples of $X$ and $Y$, respectively.

Similar to [22], MIs between node $Z$ and each of the other nodes depicted in Figure 4 have been calculated and used as a key metric to determine the mutual importance between pairs of variables.

While MI measures the impact of variables, demographic parity corresponds to the mathematical definition of fairness. Although, in most cases, *demographic parity* is used to establish the amount of bias hidden in a machine learning classifier; in this case, the metric will be used to discuss the amount of bias in the generated synthetic datasets. The definition of DP is as follows. Given a binary sensitive attribute $X$ with possible values $0/1$ and a node $Y$, a dataset is fair, i.e., does not discriminate with respect to attribute $X$, if $p(Y|X = 0) \simeq p(Y|X = 1)$. This metric will be later used to demonstrate how changing the weights of the edges causes the datasets to be more or less fair.

### 4.2. Experimental Setup

For quantitatively evaluating the degree of bias and fairness in generated datasets, a Python prototype was developed. This prototype generates the dataset according to the following input parameters:

1. The set of variables (i.e., nodes) that describes the dataset;
2. The set of categorical values associated with each of these variables;
3. The number of requested observations $n$;
4. The probabilistic network described in terms of the regression coefficients, i.e., the weight of the edges (zero if the edge is missing);
5. The value of the conditional variance for each node.

For all of the datasetes, the number of observations is equal to 100,000 and, for each categorical variable, the cutoffs have been determined to guarantee the same marginal probability for all its values. In addition, all conditional variances are equal to one.

The prototype takes advantage of numpy [23] and scikit-learn [24] packages to perform algebraic and statistical operations. In particular, once the variance–covariance matrix $\Sigma$ has been derived from the input parameters, the `multivariate_normal` function provided by numpy has been used for sampling the multivariate latent distribution. Moreover, the developed prototype exploits the implementation provided by sklearn to evaluate the mutual information among variables.

### 4.3. Impact of the Edge Weights

In the first round of experiments, the impact of the edge weights on the mutual dependencies between nodes has been studied with reference to the network in Figure 4. In particular, the weight of each edge has been varied in isolation from its default value of one to values ranging from zero to five. A synthetic dataset has been generated for each of these values for each edge. The plots reported in Figures 5 and 6 show the mutual information between the target node $Z$ and nodes $A$, $B$, $C$, $D$, $E$ as a function of edge weights. In particular, the left column of plots in Figure 5 shows the variation in edges connected to node $E$, whereas the right column shows the ones connected to node $C$. Figure 6 shows the plots varying in edge $A$-$B$.

By looking at the plots, the most apparent implications of increasing the edge weight is that the nodes connected by that edge undergo an increment of mutual information; see, for example, the MI $(D, Z)$ and $(E, Z)$ in the top left chart in Figure 5. The increase of mutual information between a node and the target node $Z$ indicates that the values assumed by it will more decisively impact the values of the target node, that is, the node will be more *important* with respect to the other nodes. A rise in importance for some nodes also has implications for other nodes, i.e., as the weight of the edge $D$-$E$ increases, nodes $D$ and $E$ become the most important nodes in the network, whereas others fall off (see the top left chart in Figure 5). On the other hand, as shown in the bottom right chart in Figure 5, the edge $C$-$Z$ shows an increment of importance for nodes $C$ (expected) and nodes $A$ and $B$, which are not connected by the interested edge. This happens because $A$ and $B$ are directly connected to $C$, so their influence on $Z$ grows as the importance of $C$ grows. Similar effects can be observed in the other charts of the figure.

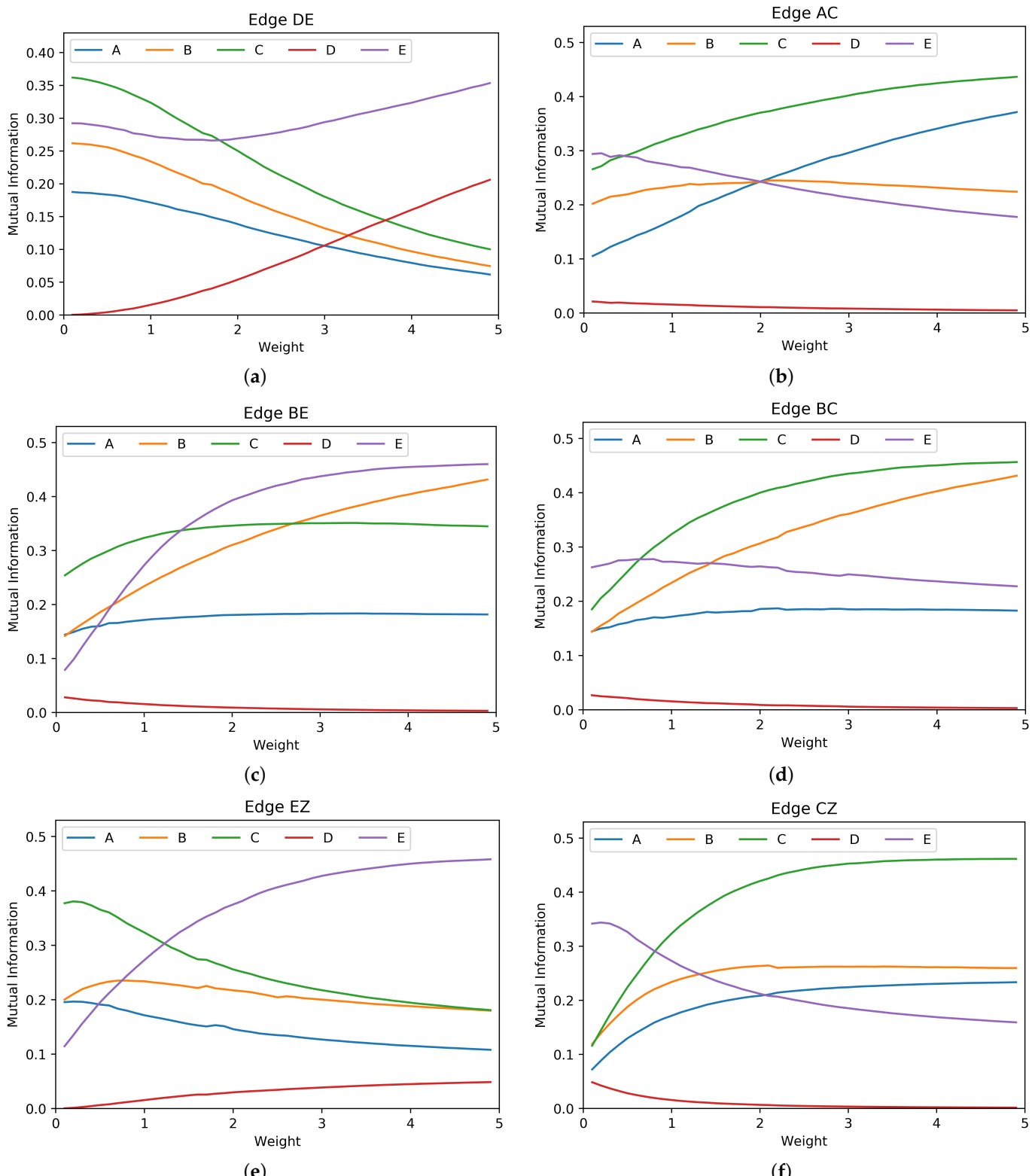

**Figure 5.** Mutual information between node Z and each other node, varying edges weight. (**a**) M.I. as function of edge DE weight. (**b**) M.I. as function of edge AC weight. (**c**) M.I. as function of edge BE weight. (**d**) M.I. as function of edge BC weight. (**e**) M.I. as function of edge EZ weight. (**f**) M.I. as function of edge CZ weight.

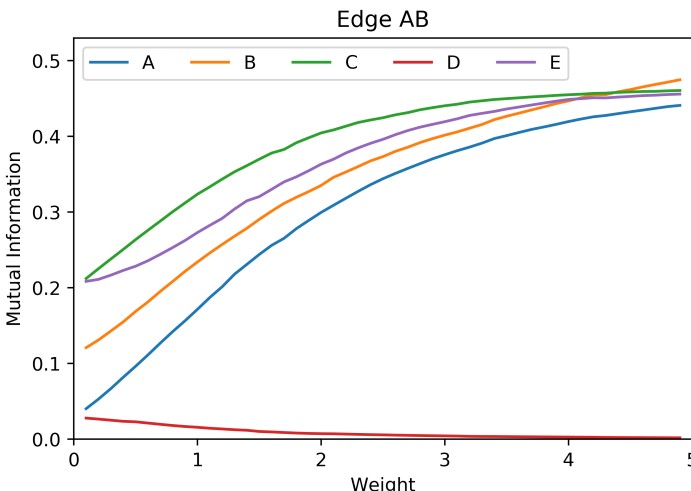

**Figure 6.** Mutual information between node $Z$ and each other node, varying the edge $A$-$B$ weight.

An interesting yet reasonable result is reported in Figure 6, where $A$-$B$ is the edge of interest: in this case, the MI significantly grows for all the nodes, except from $D$. Nodes $A$ and $B$ are direct or indirect parents to all of the other nodes except $D$; therefore, it is reasonable that increasing their connection also increases the importance of nodes $C$ and $E$. Node $D$, which is independent of $A$ and $B$, is almost not affected by a variation in the edge $A$-$B$. Interestingly, MI $(C, Z)$ grows faster than MI $(E, Z)$. This is due to the fact that $C$ is directly influenced by both $A$ and $B$; thus, increasing the importance of these nodes rapidly increases the importance of $C$. On the contrary, node $E$ is directly influenced by $B$ and $D$ (which is the unaffected node), so its importance still grows, but not as fast as $C$.

In summary, it is possible to see that increasing the edge weight of a given node increases its importance in the network, exposing it to more significant risks of bias.

### 4.4. Impact of Conditional Variance

In addition to the weight of the edges, the proposed methodology allows the user to change the overall impact of the attributes with respect to the other by varying the conditional variance of each node. The diagrams depicted in Figure 7 describe the mutual information as a function of the conditional variance of latent variables. Note that increasing the conditional variance for variables that are not connected to node $Z$ results in a gain in terms of mutual information. Figure 7 shows *MI* between node $Z$ and all of the others as a function of the conditional variance of nodes $A$, $B$ and of nodes $D$ (left plot) and $C$, $E$, $Z$ (right plot).

As expected, increasing the conditional variance of variables $A$, $B$ and $D$ results in a larger spreading of values for these latent variables, which, in turn, leads to significant increases in *MI* for nodes $C$ and $E$. This happens because the impact of a variable characterized by a flat probability distribution tends to be less relevant with respect to the other variables. Indeed, its contribution to the sum of latent variables inside each node does not significantly affect the shape of the probability distribution and, hence, the impact of other variables becomes more important. Conversely, in the right plot, the variances of nodes $C$, $E$ and $Z$ are increased and, thus, in general, all variables become less relevant.

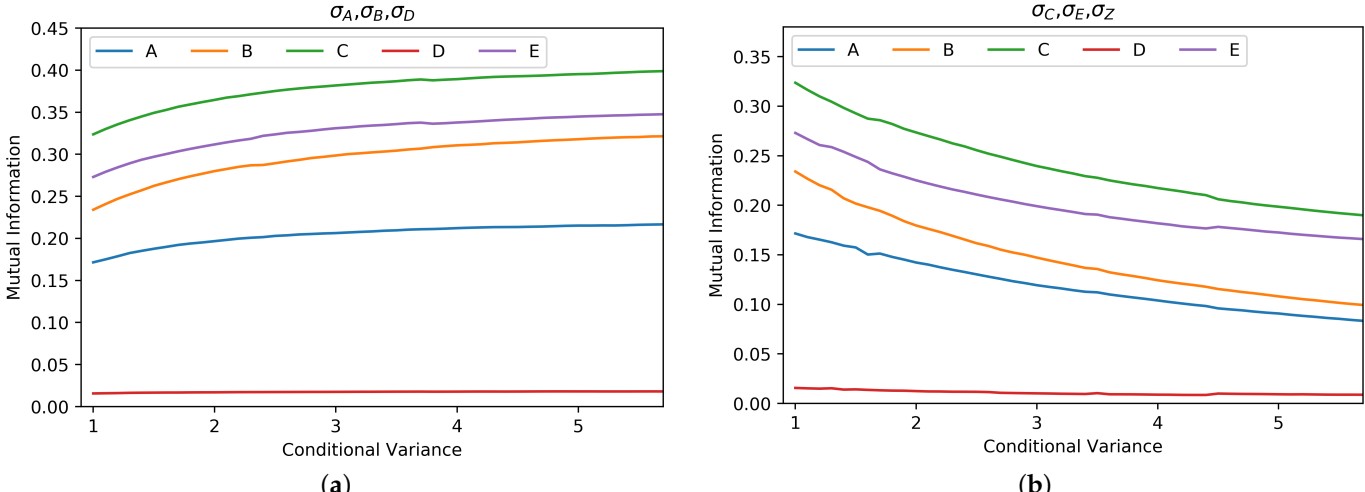

**Figure 7.** Mutual information between node $Z$ and each other node, varying conditional variances. (**a**) M.I. as function of variances $\sigma_A$, $\sigma_B$, $\sigma_D$; edge weight equal to 0.1. (**b**) M.I. as function of variances $\sigma_C$, $\sigma_E$, $\sigma_Z$; edge weight equal to 10.

### 4.5. Example of Dataset with Discrimination

The last aspect presented in this paper focuses on generating fair or unfair datasets. In what follows, an unbiased dataset and a biased dataset are generated starting from the loan approval probabilistic network shown in Figure 3. Such datasets can be used for classification tasks; in fact, as discussed in Section 3.1, the "Loan Status" attribute is binary and reflects the granting or not of a loan, given the other attributes. Since this network contains sensitive attributes such as "Gender" or "Ethnicity", it is reasonable to believe that a fair dataset, with respect to these variables, would be preferable over a discriminatory one. In the case of the attribute "Gender", for example, that would mean that the amount of positive outcomes given the gender "Female" should be equal to the amount of positive outcomes given the gender "Male". Figure 8 summarizes the principal differences between the two datasets. The top half of Figure 8 reports the histogram of attribute "Gender" and the correlation heat map of a fair dataset generated with the proposed methodology. In this case, in fact, the outcome "Loan Status" is almost not affected by the value of the attribute "Gender". These results are obtained by setting a small edges weight to all nodes connected to the attribute "Discriminatory Factor" (see Figure 3). Since the edges weight is small, the correlation between variables decreases as well, leading to a more diversified dataset.

On the contrary, the bottom half of Figure 8 shows the opposite scenario: in this case, the starting network is the same but the generated dataset is built by strengthening the bond between sensitive attributes.

Increasing these edge weights increases the discrimination towards those records including a "Female" value, with the consequent potential risk of bias when training an algorithm on this dataset. Indeed, as can be noticed from the figure, there are very different conditional frequencies for the loan approval status depending on the attribute gender, even if the corresponding correlation matrices do not highlight very large correlations between sensitive attributes and loan approval.

Generating unfair datasets can be very helpful to evaluate the effectiveness of bias mitigation approaches when developing machine-learning-based models and classifiers. Indeed, depending on the specific purpose, the proposed methodology can be exploited to generate fair or unfair datasets starting from the same network, i.e., a set of attributes, providing a large scale of options to those in need of a synthetic dataset that reflects specific correlations and patterns between observed variables.

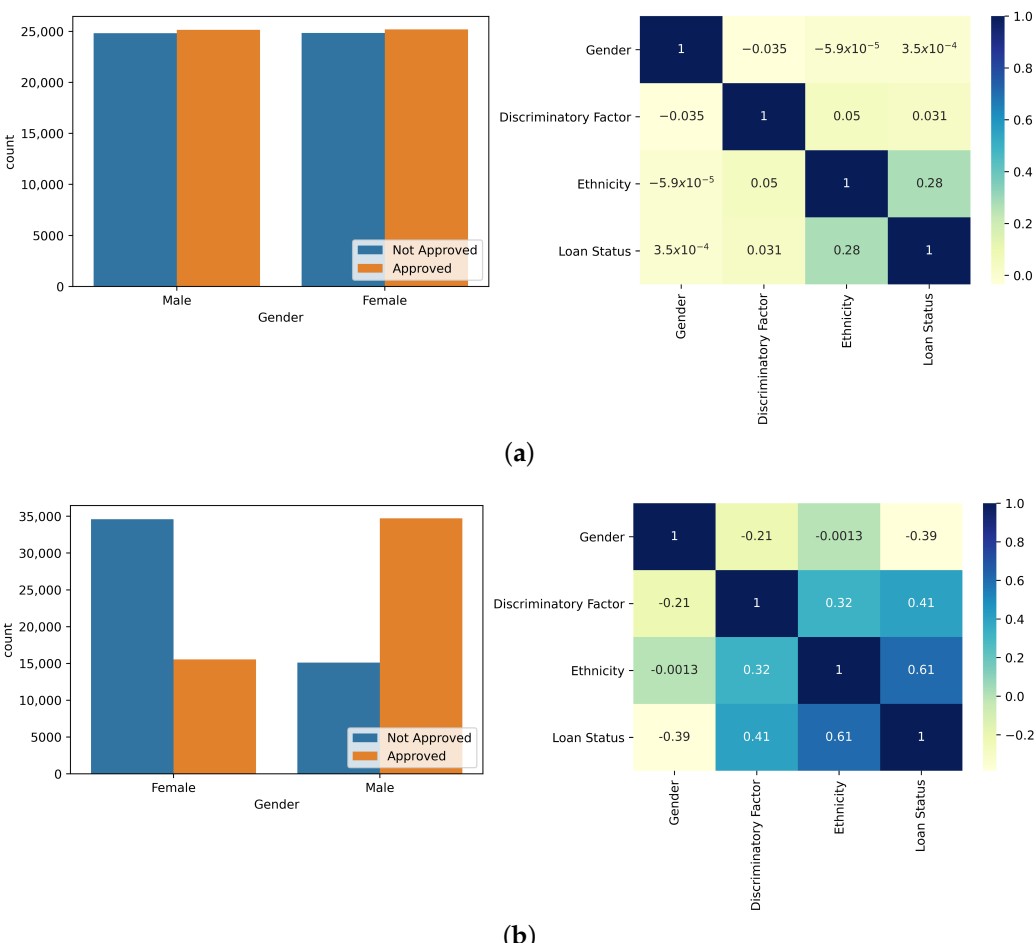

(**a**)

(**b**)

**Figure 8.** Impact of edge weight for nodes connected to node "Discriminatory Factor": when the nodes are weakly connected (top half), the generated dataset is fair with regard to attribute "Gender". Vice versa, when nodes are strongly connected, demographic parity is not respected and the generated dataset discriminates the "Female". (**a**) Gender histogram and respective heat-map of attributes connected to it with edge weight equal to 0.1. (**b**) Gender histogram and respective heat-map of attributes connected to it with edge weight equal to 10.

## 5. Conclusions

This work has presented a methodology for generating synthetic datasets with a controlled amount of bias and fairness based on a limited number of parameters. In detail, starting from a probabilistic network, the methodology allows data scientists to control both the absolute relevance of individual features and the amount of influence among connected features in generated datasets. As a proof of concept, a simple synthetic dataset consisting of six features has been generated to evaluate the impact, in terms of mutual information between pairs of features, of varying the importance of individual nodes and edges. In addition, the methodology has been used to generate a more realistic use case, based on a loan approval status dataset, to evaluate the impact of a controlled amount of discrimination, due to sensitive attributes, such as gender and ethnicity, on a target attribute, that is, the loan approval.

Controlling the amount of bias and fairness in synthetic datasets can be exploited to develop and evaluate the effectiveness of bias mitigation approaches when developing machine-learning-based models and classifiers. The proposed methodology complies with this requirement by providing a large scale of options to match specific correlations and patterns among features starting from a limited number of parameters. Future developments are expected to address the creation of an integrated tool for easing the generation

of synthetic datasets and, hence, allowing data scientists to test discrimination mitigation techniques on machine learning applications.

**Author Contributions:** Writing—original draft: M.L.D.V., D.T. (Daniele Tessera), N.V.; writing—review and editing: E.B., D.T. (Daniele Toti). All authors have read and agreed to the published version of the manuscript.

**Funding:** This work is partially supported by the Catholic University of the Sacred Heart, with grant D.3.2 "Ethic and scientific impacts of AI applications".

**Institutional Review Board Statement:** Not applicable.

**Informed Consent Statement:** Not applicable.

**Acknowledgments:** All the authors would like to thank Federico Castelletti for fruitful discussions related to structural equation modeling.

**Conflicts of Interest:** The authors declare no conflict of interest.

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
