# Peer review of "A Methodology for Controlling Bias and Fairness in Synthetic Data Generation"

_applsci, doi:10.3390/app12094619_

Round 1

Reviewer 1 Report

To systematically generate synthetic problems for model training is meaningful.  Real-world data are usually expensive and difficult to get.  This paper proposes a meaningful methodology for generating both ‘biased’ and ‘unbiased’ datasets.  The paper is overall well written, with a few concerns below that may hopefully improve the paper:

  1. Both ‘biased’ and ‘unbiased’ should be clearly specified under different context, e.g., ‘unbiased’ could refer to the equal numbers of samples, or equal run times. Generally, the term ‘biased’ has a significantly wider range than ‘unbiased’.  For example, we may consider ‘uniformly distributed difficulty’ to be ‘unbiased’, and all the non-uniform cases will be ‘biased’.  Therefore, it would be better if some suggestions and/or guidance could be provided to let the readers better understand ‘bias’ and how to add ‘bias’ onto their problems, and more importantly, is there any bad results if biases are inappropriately imposed.
  2. The possibility of using non-Gaussian noise should be introduced, and the pros and cons could be also discussed.
  3. Some related articles for synthetic problem generation are listed for consideration: https://doi.org/10.1145/3205651.3208257, https://doi.org/10.1145/3321707.3321761, https://doi.org/10.1016/j.swevo.2018.04.005

Author Response

Dear Reviewer,

please find enclosed a description of the changes made to the applsci-1655336 manuscript, in response to your comments. We thank you for the useful suggestions on how to improve the manuscript. 
We have done our best to comply with all of them. You can find the comments and a point-to-point reply in attachment.

Best regards,
Enrico Barbierato

Reviewer 2 Report

The authors present a novel method to generate a synthetic dataset, where bias can be modeled by using a probabilistic network exploiting Structural Equation Modeling.

The topic is very important in machine learning. The structure of the manuscript is also well. It should fall into Journal Appl. Sci. However, several minor questions might pay attention:

  1. In section 1, the authors gave many statements or viewpoints in context. If the authors cite some references to support your points, it will be better.
  2. In section 1, the authors could highlight your research significance as soon as possible.
  3. About equations, the authors ignored the equation’s num and give some necessary annotation for the variables. Please check it.
  4. The authors should add a sub-caption for each sub-figure in many Figures, such as Figures 5, 7, and 8.
  5. In Figure 8, the confusion matrix map is in low resolution. Please provide a high pixel image instead.
  6. The authors mentioned several data augment methods. However, there is no finding comparison between the proposed method and the previous method.
  7. The discussion also can be extended in the context.

Hopefully, this will help in the revision of the manuscript.

Author Response

(The authors gave the same response as above.)

Round 2

Reviewer 1 Report

The paper has been well revised and improved. I am happy to suggest acceptance for publication.